# The Influence of Somatic Maturity on Anthropometrics and Body Composition in Youth Soccer Players

**DOI:** 10.3390/children10111732

**Published:** 2023-10-26

**Authors:** Pavlína Kalčíková, Miroslava Přidalová

**Affiliations:** Department of Natural Sciences in Kinanthropology, Faculty of Physical Culture, Palacký University Olomouc, Křížkovského 511/8, CZ-779 00 Olomouc, Czech Republic; miroslava.pridalova@upol.cz

**Keywords:** body composition, pubescent, somatic maturity, soccer

## Abstract

The primary aim of this investigation was to compare players’ anthropometric (AP) and body composition (BC) characteristics between distinct maturity bands (pre-PHV, circa-PHV, post-PHV) in youth elite soccer. This study considered 320 male soccer players (mean age 13.8 y). Participants were from U14 (*n* = 157) and U15 (*n* = 163) age categories. The Khamis–Roche method was applied to calculate the percentage of predicted adult height (PAH) at the time of assessment based on which the players were further divided into maturity bands (pre-PHV ≤ 87%, circa-PHV = 88–95%, post-PHV > 95%). The findings indicated that most of the players were in the circa-PHV stage at the time of investigation. Measurements included height and weight. The estimates of BC parameters were derived using bioelectrical impedance (BIA) analysis. These various AP and BC attributes displayed significant differences among the maturity bands (F = 139.344–7.925; *p* < 0.001; large effect sizes) except in body fat mass (BFM) (F = 2.998; *p* = 0.051; small effect size). The current somatic maturity stage of the athletes should be considered when evaluating BC results, otherwise there is a risk of misinterpretation.

## 1. Introduction

Talent identification in youth soccer is becoming increasingly important for clubs and national associations [1]. Predicting a young player’s potential for future success at the elite-level is difficult due to its non-linearity and multi-factoriality [2]. Finding, recruiting, and nurturing talent which will be retained in the system is a global challenge [3]. The data suggests that the recurrent process of selection and deselection of youth soccer players during childhood and adolescence is responsible for the emergence of players in professional competitions more than the longitudinal continuous process of talents identified early [4].

Anthropometrical and physiological factors are among the most frequently researched topics in the talent identification area [1]. Both are significantly influenced by age, growth rate and maturation and their interactions throughout childhood and adolescence [5,6]. Maturity was previously recognized as a factor influencing the players performance and selection process in youth soccer [7,8]. Boys who are somatically accelerated have an advantage in size, strength, and power as opposed to peers who are on-time or late-maturing [9,10,11]. These maturity associated differences are most prominent between 11 and 14 years of age [10]. At about age 14, boys maturing on-time reach their peak height velocity (PHV). This is the moment when the maximum rate of growth in height occurs [12]. During this period, the natural increase in height and weight is connected to the highest rate of increments in motor skills such as sprinting, jumping, or throwing [13]. For practical use, it is possible to group players as being pre-, circa-, and post-PHV. The data from longitudinal studies indicates, that between 90% [14] and 92% [15,16] of predicted adult height (PAH), is when the PHV occurs. The growth spurt lasts about 24–26 months [17], which means we can expect the onset to be one year before the PHV (88–89% PAH) and the end of the rapid growth increments one year after the PHV (95–96% PAH) [18]. The moment when physical and functional changes are triggered vary in each individual [19]. In sports, where contact plays a crucial role, late-maturing players may be overlooked or deselected from the team due to their inherent disadvantages [19,20]. Coaches and other staff working with children and youths should take individual characteristics of the adolescents into account [13].

BC carries out information about health and functional status, and serves as a basis for nutritional recommendations, and for assessing the effectiveness of training interventions [21]. Every method for estimating BC has its pros and cons [22]. BIA is one of the safe noninvasive methods favored for its simplicity, portability and low costs. In comparison to dual energy X-ray absorptiometry (DEXA), which is considered to be the golden standard in BC estimation [23,24], BIA still provides reliable results [25,26,27]. It is important to be aware that BIA tends to underestimate percentage of body fat (%BF) and BFM and overestimates fat free mass (FFM) [27,28,29]. Due to the conductive properties of human tissues, BIA results depend on the hydration status of the individual [30]. Failure to follow protocol and measurement principles can lead to errors [31].

Previous studies related AP (height, weight, body mass index (BMI)) and BC characteristics to age [32,33,34], competitive divisions [35,36,37], playing positions [32,33,35,38,39,40,41] and to maturation [34,42,43,44,45,46]. Proportions and the relationship among AP and BC characteristics change non-linearly during childhood and puberty [12,47]. Only one previous study [46] compared AP and BC characteristics based on pre-, circa- and post-PHV divisions, which from our perspective, for practical use, is most applicable. The study goal was to contrast AP and BC characteristics of pubescent professional soccer players, in age along with the PHV, and between distinct maturity bands (pre-PHV, circa-PHV, post-PHV).

## 2. Materials and Methods

### 2.1. Participants

A cross-sectional study was carried out during the 2022/2023 season. The investigation involved male youth soccer players (*n* = 320, age 13.8 y.) from eight Czech soccer academies. Players at the elite and sub-elite level from the U14 (*n* = 157) and U15 (*n* = 163) teams participated in this research. The guideline of the Declaration of Helsinki [48] was followed. All participants participated voluntarily, and their parents or legal guardians received information and signed an informed consent document. This consent had been previously approved by the local Ethics Committee of the Palacky University in Olomouc (Ethical Approval Code: 34/2023; dated 4 April 2023).

### 2.2. Anthropometry

AP measurements were acquired following the established guidelines of the International Society for the Advancement in Kinanthropometry (ISAK) [49]. Trained academy physiotherapists measured AP characteristics. A standard portable stadiometer, BSM170B (Biospace Co., Seoul, Republic of Korea) was used for measuring body height to the nearest 0.1 cm. Weight to the nearest 0.1 kg, FFM (kg), skeletal muscle mass (SMM; kg), BFM (kg), %BF (%), total body water (TBW; l), segmental FFM of trunk (FFMT; kg) lower right and left limb (FFMRL, FFMLL; kg) and upper right and left limb (FFMRA, FFMLA; kg) were measured via a BIA analyzer Inbody 270 (Biospace Co., Seoul, Republic of Korea) using an electric current at frequencies of 50 kHz and 100 kHz. The validity and reliability of using BIA for BC analysis have been proven in previous studies [26,50,51]. The measurements were carried out barefoot in only light clothes (shorts) in the morning hours on an empty stomach. Participants were standing on electrodes and holding handheld electrodes in their hands. Relevant guidelines for measurement according to InBody were followed.

### 2.3. Maturity Status

The percentage of PAH achieved at the time of measurement was used as a non-invasive estimate of somatic maturation in US and European Youths [52,53], and was proven to have satisfactory concurrent validity [54]. Even so, it is necessary to take into account that PAH is dependent on initial data, which have their own ethnic dependence [55]. The Khamis-Roche method [56] was used to predict the final adult height in youths as in previous studies [53,57,58]. This method uses the current individual’s chronological age, height, weight and calculated mid-parental height (biological parents’ average height). Between the actual and PAH, the median error in ages 4 to 18 in males is 2.2 cm [56]. The self-reporting approach was used when collecting the biological parents’ height. Self-reported heights were adjusted with an over-estimation equation [59]. Youth players can be grouped into maturity bands using the percentage of PAH [18,58,60,61]. The pre-PHV band was set to ≤87% of the adult stature, circa-PHV 88–95% of adult stature and in the post-PHV band were players with >95% of the adult stature, similarly to previous studies [62,63,64].

### 2.4. Statistical Analyses

Descriptive statistics were provided via averages and standard deviations (SD). Categorical variables (maturity status) were determined as frequencies and percentages. The disparities in the frequencies were assessed using the Chi-squared test. When significant differences in frequency counts were identified, a post hoc test was used and Cramer’s V was employed to assess the extent of the effect size (0.07 = V < 0.21—small effect, 0.21 = V < 0.35—medium effect, V ≥ 0.35—large effect) [65]. A comparison was made within three maturity bands (pre-PHV, circa-PHV, post-PHV) using a one-way analysis of variance (ANOVA). In cases where ANOVA revealed a notable influence related to maturity bands, the Bonferonni post-hoc test was applied to assess the distinctions among separate bands. Effect size was calculated to test for the practical significance. Effect sizes were computed using partial eta squared (η^2^) and were assessed based on Cohen’s criteria for interpretation (0.01 = small, 0.06 = medium, and 0.14 = large) [65]. The data was processed with the SPSS program version 23 (SPSS Inc., Chicago, IL, USA). The significance level was established at *p* < 0.05.

## 3. Results

The mean age of the sample was 13.8 ± 0.6 years (range: 12.7–14.7 years), height 167.7 ± 9.5 cm (range: 145.4–190.8 cm) and weight 53.3 ± 9.7 kg (range: 31.5–84.4 kg). Frequency counts and percentages of participants in maturity bands can be seen in Table 1.

Most of the players were identified as circa-PHV (76.3%). A significantly lower rate of post-PHV players was identified in the U14 (1.3%) age group compared to the U15 (25.8%) age group. Conversely, in the U15 (1.2%) age group, there was a lower rate of pre-PHV players compared to the U14 (19.1%) age group. The frequency counts of players in pre-PHV and post-PHV bands differed significantly within age-groups (U14, U15) X^2^ (2, *N* = 320) = 78.778, *p* < 0.001, Cramer’s V = 0.496, large effect. AP and BC characteristics according to maturity status (pre-PHV, circa-PHV, post-PHV) are shown in Table 2.

As players matured, the mean values of all characteristics increased, except for the %BF which on the contrary, decreased. Significant differences within maturity bands were identified in all observed AP and BC characteristics (F = 139.344–7.925; *p* < 0.001; large effect sizes) except in BFM (F = 2.998; *p* = 0.051; small effect size). After the Bonferroni post-hoc test adjustment, significant differences were identified between bands in all AP and BC variables (*p* < 0.001), except in the BFM and %BF. The %BF differed between the most mature (post-PHV) and both other bands (*p* = 0.001–0.005). BFM differed only between pre-PHV and the post-PHV band (*p* = 0.046), even if the overall difference between bands were not identified. The ANOVA results, F values, the main effect, and comparison after the Bonferroni pairwise adjustment among the maturity bands can be seen in Table 3.

## 4. Discussion

The aim of this study was to investigate how somatic maturity in young soccer players impacted AP and BC characteristics. ANOVA demonstrated differences in the three maturity bands (pre-PHV, circa-PHV, post-PHV) in all examined parameters except the BFM. The development of %BF in our study behaved differently from other BC characteristics. The more mature the players were, the higher the AP and BC characteristics. Only the %BF decreased during maturation, contrary to the rest of the observed variables. In addition, we found that the two age-groups (U14, U15) differed in players’ frequency counts in the distinct maturity bands. The overall results showed that the maturity status of most of the players at the time of observation was circa-PHV (76.3%). According to our results the AP and BC characteristics favor those athletes who already reached PHV.

Studies that investigate BC in youth athletes in diverse maturity stages are quite rare. Comparison of this study with previous studies is complicated because of the use of different methods for BC estimation. Also, methods used for maturity status determination differ. Nevertheless, the results of this investigation are consistent with previous studies in some conclusions [42,45,46,66], where AP and/or BC characteristics in athletes within different maturity stages were also observed.

In our study it was detected that all maturity bands differed significantly between each other in height, weight, and BMI. These conclusions are consistent with the studies of Di Credico et al. [46], Albaladejo-Saura et al. [42] and Toselli et al. [45] who also compared AP characteristics within distinct maturity bands in youth athletes. It is necessary to note that only the Di Credico et al. [46] study used the same three maturity bands as in our study, even though a different method to set the maturity status was used. Contrary to our results, in the Di Credico et al. [46] study the difference in weight and BMI was found only between pre-PHV and the two remaining bands. This could be explained by the discrepancy in weight in circa-PHV bands, which in our study was lower (current study weight = 51.9 ± 7.6 kg, Di Credico et al. [46] weight = 62.7 ± 10.2 kg) while the observed height was similar in both studies (current study height = 166.6 ± 7.6 cm, Di Credico et al. [46] height 166.4 ± 7.6 cm). Similar to our method, Albaladejo-Saura et al. [42] followed two circa-PHV bands (age at PHV = 0.1–0.5) and one band in the post-PHV stage (age at PHV = 1.6). This study demonstrated differences between post-PHV and both circa-PHV bands in height, weight, and BMI, but did not confirm the same conclusions between the two circa-PHV bands in height and weight. Furthermore, Toselli et al. [45] compared AP and BC characteristics within three maturity bands where from our perspective one was circa-PHV (age at PHV = −0.6) and two were pre-PHV (age at PHV < −1.3). In the Tosselli et al. [45] study the difference was found in all AP characteristics between the circa-PHV band and the two pre-PHV bands but not between the two pre-PHV bands. Differences in AP characteristics in distinct maturity stages could be explained by hormonal changes in adolescence [12]. The hormonal process of controlling maturation is a very complex process and detailed discussion of hormones related to maturation is beyond the scope of this study.

Additionally, the study found that there is an increase in the amount of FFM and TBW with maturation and observed differences between distinct maturity bands which are therefore in line with the Toselli et al. [45] conclusions. These results underpin the findings of Campa et al. [43], where FFM showed asynchronous development, which significantly increased 2.2 years before PHV and disappeared post-PHV (1.3 years after PHV). The difference among distinct maturity bands was in this study also in SMM and the same conclusions can be also found in the Albaladejo-Saura et al. [42] study.

According to our results and previous conclusions [12,45,46] the volume of BFM increased progressively as the players become more mature. Only in the Di Credico et al. [46] study was the highest amount of BFM in the circa-PHV group. In this study there was no significant difference in BFM. Di Credico et al. [46] and Toselli et al. [45] found a difference between pre-PHV and circa-PHV bands and Albaladejo-Saura et al. [42] noticed a difference between the post-PHV band and the two-remaining bands (both circa-PHV). In our study the tendency of maturation leading to an increase in BFM and a decrease in %BF was present. Our results agree only with the Di Credico et al. [46] study in this regard. Opposite results are presented in some previous studies [45,66,67], where the more mature the athletes the higher the %BF was found. Malina and Bouchard state that the highest %BF is at around 11 years of age and then decreases until late puberty when after that %BF remains relatively constant. The decrease in %BF during male growth and maturation is explained by the study from Malina et al. [47] This study explains the influence of the accelerated growth of FFM and the slower rise of BFM and that is why the proportional contribution of BFM is lower in male adolescents. The result of the current investigation presents a difference in %BF in maturity bands only between post-PHV and the remaining maturity bands (pre-PHV, circa-PHV). In this direction Di Credico et al. [46] found the difference in distinct maturity bands as well, but unlike our study, among all the maturity bands. Toselli et al. [45] and Rusek et al. [34] found no difference between maturity bands in %BF. In the case of the amount of BFM and %BF, a different method of determining the total amount of fat tissue can play a significant role. In studies using the BIA method there was a lower level of BFM (5.1–11.4 kg) in all maturity bands compared to other studies (8.1–19.8 kg). This can be explained by the tendency of BIA to underestimate BFM and %BF and overestimate FFM which was previously well described [27,28,29,68]. Also, nutrition and the training process, make the BFM and %BF more variable [47], a different kind of sport or specific position (goal keepers) may also contribute to differences in results.

The discrepancy in the results between current and previous research, except the already mentioned distinctions, may also be caused by the age range of the participants. As we examined only the players from U14 and U15 categories (range: 12.7–14.7 years) where player’s age and maturity status aligned closely to the PHV, the results may present different conclusions than studies where the span of the participants’ age is wider (i.e., 9–18 years). Therefore, during puberty, the use of annual age categories would be a suitable solution.

Additionally, a different approach (late-, on-time-, early maturers vs. pre-, circa-, post-PHV players) used for maturity status estimation can lead to inconsistent results. Early maturing players and late maturing players are usually defined as those whose calendar age (CA) differs from the estimated ±1 year [69]. This construct refers to a lag in age as opposed to pre-, circa-, and post-PHV, which refers to how far (±1 year) from PHV an individual is at the time of observation. It follows that an early maturing player at age 10 may still be pre-PHV, and on the other hand, a late-maturing 15-year-old player may also still be pre-PHV. This discrepancy highlights the need to consider whether and how the researchers define maturity status.

In practical use, based on current and previous research evaluation of AP and BC characterization of players going through adolescence should be made with consideration of an actual maturity status. Conclusions about BFM and %BF especially, should be made with all due respect to the natural non-linear development of these variables and with awareness that no values for optimum BC for soccer players exist. It always depends on an individual player’s physiology, field position and/or playing style [70]. From the performance perspective, maturity related (dis)advantages have been reported within distinct maturity bands [71,72,73]. Other factors such as AP and BC parameters can contribute positively or negatively to performance in youth athletes [74,75]. Coaches and other professionals working with young athletes should be aware that current differences in performance may be caused by a discrepancy in the timing of physical maturation and the associated development of individual body parameters and attributes of body composition.

Furthermore, the sample division according to maturity status showed that overall, about three-fourths (76.3%) of the boys in U14 and U15 categories were circa-PHV. In accordance with previous findings, this period in their life can present “adolescent awkwardness” [13], worse recovery capabilities [76], the risk of growth-related injuries [77,78,79,80] and the overall injury incidence for circa-PHV players is higher compared to pre-PHV players. Also, the physiological response to training differs within maturity bands [76]. Based on these findings particularly for those players going through a growth spurt an individual approach is necessary and applicable strategies are already set [81]. A positive contribution of this study is that among the few that investigated the influence of maturation to BC, this study includes both AP and BC characteristics and all maturity stages (pre-, circa-, post-PHV).

Also, in this study some limitations need to be considered. First, the anthropometric measurements were conducted by various examiners. Secondly, the number of circa-PHV players in the present study is larger than the number of players in the two other bands, which may influence the statistics and thus also the results. Another weakness of the study is not controlling for possible confounding effects of CA or previous training exposure, which may have improved the results. Finally, even though the Khamis–Roche method [56] is widely used for determining somatic maturation and its use is reasonable when evaluating hundreds of participants, it has lower accuracy than the “gold-standard” [82]. Another limit associated with the Khamis–Roche method is the use of self-reported parents’ adult heights. Using two alternative methods concurrently would yield more precise results. Knowing the actual growth rate would also be valuable information.

## 5. Conclusions

The research demonstrates that in the period of pubescence, the level of somatic maturation influences AP and BC characteristics, except for the BFM in male youths playing soccer. As players became physically mature, the values of their body parameters increased, only %BF showed the opposite trend. Therefore, coaches and other staff working with children and youth should understand how differences in maturity status can influence not only AP and BC attributes, but all the connected aspects of the player’s development. It is important for the player’s health, development and also in the process of (de)selection that the current level of growth rate and somatic maturation are regularly determined and taken into account when interpreting results and drawing conclusions. Windows of opportunity open for each child/talent at a different time of their development.

Furthermore, in the Czech soccer context, this study found that most of the players in U14 and U15 categories are going through their growth spurt which should be considered when drawing up load planning in the short and long-term.

## Figures and Tables

**Table 1 children-10-01732-t001:** Statistics of frequencies in categorical variables.

Variables	ALL (*n* = 320)	U15 (*n* = 163)	U14 (*n* = 157)
Pre-PHV	32 (10.0%)	2 (1.2%)	30 (19.1%)
Circa-PHV	244 (76.3%)	119 (73.0%)	125 (79.6%)
Post-PHV	44 (13.8%)	42 (25.8%)	2 (1.3%)

Pre-PHV ≤ 87% of adult stature; Circa-PHV = 88–95% of adult stature; Post-PHV > 95% of adult stature.

**Table 2 children-10-01732-t002:** Descriptive statistics of AP and BC variables in distinct maturity bands.

	All (*n* = 320)	Pre-PHV (*n* = 32)	Circa-PHV (*n* = 215)	Post-PHV (*n* = 73)
Variables	Mean (SD)	Mean (SD)	Mean (SD)	Mean (SD)
Age (years)	13.82 (0.57)	13.17 (0.30)	13.74 (0.52)	13.82 (0.57)
%PAH (%)	91.72 (3.21)	86.09 (0.93)	91.14 (1.99)	95.88 (0.96)
Height (cm)	167.70 (9.54)	153.32 (4.74)	166.60 (7.56)	177.23 (6.00)
Weight (kg)	53.29 (9.71)	40.42 (3.86)	51.85 (7.63)	63.18 (7.72)
BMI (Kg/m^2^)	18.79 (1.88)	17.18 (1.26)	18.59 (1.68)	20.08 (1.92)
TBW (L)	35.03 (6.61)	26.12 (2.45)	33.98 (5.22)	42.00 (4.73)
BFM (kg)	9.45 (1.80)	7.03 (0.65)	9.17 (1.43)	11.36 (1.28)
%BF (%)	10.51 (3.53)	11.91 (4.05)	10.74 (3.49)	9.25 (3.05)
FFM (kg)	47.72 (9.04)	35.55 (3.30)	46.30 (7.14)	57.24 (6.47)
SMM (kg)	26.51 (5.45)	19.21 (1.97)	25.64 (4.30)	32.27 (3.91)
FFMRA (kg)	2.37 (0.65)	1.55 (0.26)	2.27 (0.52)	3.05 (0.49)
FFMLA (kg)	2.35 (0.65)	1.54 (0.23)	2.24 (0.51)	3.04 (0.49)
FFMRL (kg)	7.74 (1.71)	5.37 (0.63)	7.45 (1.35)	9.55 (1.16)
FFMLL (kg)	7.68 (1.69)	5.32 (0.60)	7.42 (1.34)	9.47 (1.13)
FFMT (kg)	20.38 (3.99)	15.09 (1.52)	19.72 (3.15)	24.62 (2.84)

%PAH = percentage of predicted adult height, TBW = total body water, BFM = body fat mass, %BF = percentage of body fat, FFM = fat free mass, BMI = body mass index, FFMRA = fat free mass right arm, FFMLA = fat free mass left arm, FFMRL = fat free mass right leg, FFMLL = fat free mass left leg, FFMT = fat free mass trunk.

**Table 3 children-10-01732-t003:** Difference in AP and BC variables in distinct maturity bands.

Variables	F Value	ANOVA	Effect Size		Bonferroni Post-Hoc Test	
	F	*p*	η^2^	Pre-PHV– Circa-PHV	Circa-PHV– Post-PHV	Pre-PHV– Post-PHV
Age (years)	77.601	<0.001	0.329 (large)	<0.001	<0.001	<0.001
%PAH (%)	394.183	<0.001	0.713 (large)	<0.001	<0.001	<0.001
Height (cm)	138.056	<0.001	0.466 (large)	<0.001	<0.001	<0.001
Weight (kg)	118.743	<0.001	0.428 (large)	<0.001	<0.001	<0.001
BMI (kg/m^2^)	36.578	<0.001	0.188 (large)	<0.001	<0.001	<0.001
TBW (L)	131.383	<0.001	0.453 (large)	<0.001	<0.001	<0.001
BFM (kg)	2.998	0.051	0.019 (small)	0.247	0.508	0.046
%BF (%)	7.925	<0.001	0.048 (small)	0.223	0.005	0.001
FFM (kg)	131.134	<0.001	0.453 (large)	<0.001	<0.001	<0.001
SMM (kg)	131.537	<0.001	0.454 (large)	<0.001	<0.001	<0.001
FFMRA (kg)	119.185	<0.001	0.429 (large)	<0.001	<0.001	<0.001
FFMLA (kg)	124.524	<0.001	0.440 (large)	<0.001	<0.001	<0.001
FFMRL (kg)	137.366	<0.001	0.464 (large)	<0.001	<0.001	<0.001
FFMLL (kg)	139.344	<0.001	0.468 (large)	<0.001	<0.001	<0.001
FFMT (kg)	131.611	<0.001	0.454 (large)	<0.001	<0.001	<0.001

%PAH = percentage of predicted adult height, TBW = total body water, BFM = body fat mass, %BF = percentage of body fat, FFM = fat free mass, BMI = body mass index, FFMRA = fat free mass right arm, FFMLA = fat free mass left arm, FFMRL = fat free mass right leg, FFMLL = fat free mass left leg, FFMT = fat free mass trunk.

## Data Availability

The data presented in this study is available upon request from the respective authors. Due to the protection of personal data, the data is not publicly available.

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
