# Peer review of "The Influence of Somatic Maturity on Anthropometrics and Body Composition in Youth Soccer Players"

_children, 2023, doi:10.3390/children10111732_

Round 1
Reviewer 1 Report
The search for athletically gifted youth is important for sports clubs and is an important factor in the subsequent education and training of high-performing athletes. This also applies to team sports such as football.
The article is an attempt to answer the question of what the anthropometric features and body structure are and whether there are significant differences among male youth playing football in the Czech Republic between stages of maturity (before PHV, around PHV, after PHV).
Taking into account the fact that in the case of young athletes, the expected age of maturity and the age of maximum growth velocity are increasingly used for their qualification, the aim of the article was to compare the anthropometric characteristics and body composition of athletes belonging to elite young football players in the Czech Republic at different periods of maturity (before PHV , around PHV, after PHV). While the results of anthropometric tests of athletes are published quite often, they rarely concern the impact of growth intensity on the values of anthropometric indicators. In the available literature, the reviewer did not find any works on the topic of the population of young Czech footballers discussed in the article, which should be treated as an innovative element.
The authors showed that the age-related degree of somatic maturity of young football players affects anthropometric characteristics and body composition in three ways (before PHV, around PHV, after PHV) for all health parameters except fat tissue mass.
The strengths of the presented work include both its cognitive value and the possibility of using the obtained results in practice to develop training programs for specific age groups, as well as appropriate nutrition models affecting the fitness and nutritional status of young football players.
Although the conclusions drawn by the authors are correct and result from the analysis of the obtained research results and contemporary literature on the subject and constitute a comprehensive answer to the questions contained in the work, there are too many of them and they are too detailed. According to the reviewer, 3-4 generalized conclusions should be provided.
The literature is contemporary, well selected and correctly cited in the text.
Tables presented in the publication are well constructed, legible and fully take into account the data contained in the content of the work.
Author Response
Dear Reviewer,
thank you for your comments. Responses to your comments can be found in the attachment. Please see the attachment.
PK

Reviewer 2 Report
The aim of this study was to compare anthropometric and body composition characteristics among young elite football players at different stages of maturation (pre-peak height velocity, around peak height velocity, post-peak height velocity). The study included 320 young male football players with a mean age of 13.8 years.
The Khamis-Roche method was used to calculate the percentage of predicted adult height (PAH) at the time of observation, which further categorized the players into different maturation stages. Height and weight were measured, and various body composition parameters, including body mass index (BMI), fat-free mass (FFM), total body water (TBW), body fat mass (BFM), percentage of body fat (%BF), skeletal muscle mass (SMM), FFM of both upper and lower limbs, and FFM of the trunk were estimated using In-Body 270.
All the observed anthropometric and body composition characteristics showed significant differences between maturation stages, except for body fat mass (BFM). It was found that the current somatic maturation stage of athletes should be taken into account when interpreting body composition results to avoid misinterpretation.
The study is well-written, and the research provides valuable insights into the relationships between maturation phases and body composition in young football players. The authors conducted thorough analyses and justified their conclusions with data. However, it would be beneficial to provide more context and emphasize the practical implications of these findings for coaches, selectors, and other professionals working with youth in sports. Overall, it is a robust study that can contribute to a better understanding of the complex associations between maturation and body composition among young football players.
Overall, the paper is written correctly, following the requirements and steps of the scientific method. However, I have a few comments regarding the overall quality of the work. In my opinion, the paper lacks scientific novelty and does not contribute significantly to the field. The methodology could be written more concisely and clearly, with explicit inclusion criteria. The paper has a practical character, and therefore, it would be valuable to add an application-based conclusion. Perhaps adding some illustrative figures could improve clarity. In the discussion, it is important to focus on the essence of the study's results. Based on my description, it seems that the paper can be improved, but it can also be published in its current form.
Author Response

(The authors gave the same response as above.)

Reviewer 3 Report
This paper presents a study conducted on youth Czech football players to identify any differences in anthropometric and body composition variables with somatic maturation. The topic is interesting and the sample size is large but the article has several weaknesses.
The first issue concerns poor English: I suggest a comprehensive revision of the English language throughout the manuscript because of too many errors.
Moreover, the discussion does not report the limitations of the study.
Minor concerns:
- Sport played: In the article (see title and text) soccer and football are used indifferently. I propose to homogenize the terms.
- Lines 128-129: the sentence repeats the data of the table and can be omitted.
- Line 258: explain the abbreviation “resp.”.
-Discussion: aside from integrating the limitations of the research, the discussion should be simplified as it is long and wordy.
Poor quality of English: errors in spelling, grammar, and punctuation, faulty sentence construction.
Author Response

(The authors gave the same response as above.)
